# Coordinated Radio Emitter Detection Process Using Group of Unmanned Aerial Vehicles

**DOI:** 10.3390/s25237298

**Published:** 2025-11-30

**Authors:** Maciej Mazuro, Paweł Skokowski, Jan M. Kelner

**Affiliations:** Institute of Communications Systems, Faculty of Electronics, Military University of Technology, 00-908 Warsaw, Poland; pawel.skokowski@wat.edu.pl (P.S.); jan.kelner@wat.edu.pl (J.M.K.)

**Keywords:** spectrum monitoring, unmanned aerial vehicle, data fusion, drone swarm, sensor network

## Abstract

**Highlights:**

**What are the main findings?**
Utilizing unmanned aerial vehicles (UAVs) in spectrum monitoring increases sensor range and freedom of movement compared to ground-based systems, and when deployed based on a coordinated group or swarm of UAVs allow simultaneous spectrum monitoring over a larger area, translating into increased range and coverage of the system.Implementation of data fusion algorithms enables real-time sensing and increases the reliability of multi-source data-based decision-making.

**What are the implications of the main findings?**
Groups of UAVs are becoming increasingly effective tools for spectrum monitoring and building situational awareness in the electromagnetic environment.The proposed system architecture enables scalability and adaptive solutions for electromagnetic spectrum monitoring in cognitive radio applications.

**Abstract:**

The rapid expansion of wireless communications has led to increasing demand and interference in the electromagnetic spectrum, raising the question of how to achieve reliable and adaptive monitoring in complex and dynamic environments. This study aims to investigate whether groups of unmanned aerial vehicles (UAVs) can provide an effective alternative to conventional, static spectrum monitoring systems. We propose a cooperative monitoring system in which multiple UAVs, integrated with software-defined radios (SDRs), conduct energy measurements and share their observations with a data fusion center. The fusion process is based on Dempster–Shafer theory (DST), which models uncertainty and combines partial or conflicting data from spatially distributed sensors. A simulation environment developed in MATLAB emulates UAV mobility, communication delays, and propagation effects in various swarm formations and environmental conditions. The results confirm that cooperative spectrum monitoring using UAVs with DST data fusion improves detection robustness and reduces susceptibility to noise and interference compared to single-sensor approaches. Even under challenging propagation conditions, the system maintains reliable performance, and DST fusion provides decision-supporting results. The proposed methodology demonstrates that UAV groups can serve as scalable, adaptive tools for real-time spectrum monitoring and contributes to the development of intelligent monitoring architectures in cognitive radio networks.

## 1. Introduction

In recent years, unmanned aerial vehicles (UAVs), known as drones, have drawn increasing interest in the context of modern communication systems [1,2]. Their miniaturization of components, decreased production costs [3], and widespread availability have made them a key tool in engineering applications. UAVs have found their application in many aspects of everyday life—conducting search and rescue, working as communication relays, border patrolling, and even delivering parcels. Drone usage has been most visible over the past few years, given the growth in the number of UAVs available on the market and their use for various tasks. The variety of available systems and their applications enables the development of new areas where unmanned systems have not been used before.

The increasing demand and development of wireless communications is driving the development of new solutions that leverage available technologies—artificial intelligence, quantum communication, and unmanned systems. One example of UAV use is supporting communication systems [4], due to greater mobility and altitude control compared to stationary transmitters (TXs). The use of single UAVs as well as groups and swarms requires skillful management and control of individual system elements. This poses a significant challenge in terms of physical resources such as airspace, the safety of third parties, and the reorganization of communication systems in order to ensure effective and reliable control of the entire system. Autonomous flight [5] is a prime example of a solution that requires reliable communication at every mission stage due to the potential risk to the environment and bystanders. UAVs constitute a crucial element of modern solutions, hence the extensive research on this topic. In addition to topics related to control, obstacle avoidance [6], and communication links [7], considerable attention is paid to the possibility of treating UAVs as platforms carrying a variety of sensors. These include daytime and thermal imaging cameras, standard equipment, through sensor arrays for air quality monitoring and chimney inspections, and radio sensors used for both spectrum monitoring and emission source location.

The growing number of devices using the electromagnetic spectrum has made spectrum monitoring a necessary task. Situational awareness in the electromagnetic spectrum allows for effective task planning, excludes mutual interference, and enables the efficient use of available resources. Currently, there is a visible tendency to engage single and mobile systems, including UAVs, for sensing [8,9,10]. Modern technologies increasingly exploit the electromagnetic environment, including in wireless communications and the Internet of Things (IoT), where monitoring resources is a key task, as are detecting, tracking, and classifying received signals. Detecting and locating unauthorized TXs is a fundamental challenge facing today’s communication systems.

The use of UAVs equipped with a single sensor was effective. However, the increased complexity of the analyzed spectrum and limitations due to spatial range require greater real-time analysis capabilities. In addition, traditional solutions may be unreliable when operating in an environment of strong interference and a lack of a positioning signal. As a result, there is a growing demand for systems that include more UAVs and are additionally equipped with a radio sensor that enables cooperation and coordination of cooperative spectrum monitoring.

In recent years, significant progress has been observed in developing cooperative monitoring systems [11,12], but many new solutions primarily use data fusion methods based on the hard-decision approach. They may encounter significant limitations when used in conditions characterized by high levels of interference, mutual interference, or the use of a link by an unauthorized user. In a dynamically changing environment, information remains a significant challenge. From previous research [13], we learned that data fusion based on Dempster–Shafer theory (DST) outperforms the classical approach (AND, OR rules and majority voting). It is clearly visible that for each desired $P_{fa}$ (0.01, 0.05, and 0.1), the number of signal detections is higher for the DST approach than for classical methods. In the case of the system’s higher probability of detection, the number of detected signals with the DST approach exceeded the values obtained with the hard methods by several times. Thanks to this feature, DST has been used in different IoT solutions [14] and cooperative spectrum monitoring [15]. Based on previous simulation studies [16,17], we wanted to extend the scope of our work to include the analysis of the behavior of a group of sensors in a near-real environment. To contribute to the development of this area, this paper proposes a test architecture and a simulation-based evaluation in the MATLAB R2025a programming environment of cooperative electromagnetic spectrum monitoring that uses a group of UAVs equipped with radio sensors based on software-defined radio (SDR). The system uses a data fusion model based on the DST, which allows modeling uncertainty and partial evidence from spatially distributed sensors. This approach increases detection accuracy, reduces false alarm probability, and supports building situational awareness in real operational conditions.

The contributions of this work are fourfold:Operational Background: This paper presents an overview of the evolution of UAVs in modern communication systems, highlighting their impact and use in sensing operations, such as spectrum monitoring, detection of energy, or emitter location—supported by recent examples of use.Architecture Framework: A simulation-based architecture is proposed in a software environment that will be used to evaluate the effectiveness of cooperative spectrum monitoring using a group of UAVs equipped with SDRs designed to operate in areas of strong interference.Data Fusion Approach: The system uses data fusion algorithms based on DST, which allows for modeling uncertainty and partial evidence, providing an alternative to conventional hard- (HDF) and soft-decision fusion (SDF) techniques.Performance evaluation: This paper presents the results of simulation studies in the MATLAB environment, which show that the system can improve detection accuracy, reduce the probability of false alarms, and increase situational awareness.

The rest of the paper is organized as follows. Section 2 presents the general concept of a spectrum monitoring system based on data fusion from a group of UAVs equipped with sensors. The evaluation of the effectiveness of the proposed solution based on simulation studies is shown in Section 3. The paper is summarized in Section 4.

## 2. Building Electromagnetic Situational Awareness Based on UAVs and Spectrum-Sensing Algorithms

Over the past few years, UAVs have been used for sensing, spectrum monitoring, and surveillance, but previously, UAVs were mainly used for reconnaissance with built-in optical cameras. Technological advances have allowed the expansion of UAV capabilities, with additional sensors performing tasks in different areas. Currently, various elements can be attached to UAVs—from air quality sensors, through communication relays, to cognitive radios whose task is to build situational awareness in the electromagnetic spectrum. UAVs can perform a wide range of activities depending on their purpose, area of operation, design type, and equipment limitations.

Scientific publications and reports divide the use of UAVs for tasks into two main groups: single-sensor systems and multi-element systems [18]. Single UAVs, which may be equipped with a radio sensor, e.g., SDR or other spectrum analyzer, have been tested in real-world conditions with the intention of scanning frequencies, determining emitters’ locations, or finding unauthorized access to the frequency band. The systems are used to patrol border areas, protect crucial facilities, and build spatial maps of operating radio assets. Due to their limited spatial range and lack of resistance to interference and external interference, their use in operational environments is limited.

Spectrum monitoring is evolving towards systems consisting of multiple UAVs, enabling new methods of building situational awareness in the electromagnetic spectrum, based on distributed detection [19]. These systems use the spatial deployment of a group of UAVs, which, thanks to real-time cooperation, can monitor a wider frequency band and a larger area (see Figure 1).

The main benefits of such a solution include increased range, the ability to localize the signal source using triangulation or time difference of arrival (TDoA) methods, and greater resistance to interference, jamming, and the operation of unauthorized users [20,21].

### 2.1. Utilization of Data Fusion Algorithms in Spectrum Sensing

Spectrum monitoring is fundamental to building situational awareness in the electromagnetic spectrum. The main task is to detect primary user transmissions in the analyzed frequency band and determine their characteristics and direction of origin. There are descriptions of many sensing methods available in the literature [22,23], which have been tested using SDR technology. The most popular of them are as follows:Energy detector (ED): The simplest of the spectrum-monitoring methods, which performs sensing by calculating the energy of the received signal in the analyzed frequency band and comparing it with a previously defined threshold. However, it does not allow for determining the type of modulation used and signal characteristics, and it does not require high computing power due to its simplicity [13,24]. The disadvantages of the above method include high sensitivity to high-power noise and interference, and low efficiency when working in an environment with a low signal-to-noise ratio (SNR). Increasing the number of sensors and using data fusion methods allows the system based on the energy detection algorithm to achieve better results, even in environments with low SNR values [25]. Another issue to tackle in a non-cooperative system is the shadowing problem [26]. The performance of spectrum monitoring individually by each sensor may be affected by reflection from objects like buildings. Empowering multiple sensors to conduct cooperative spectrum sensing may allow them to overcome deteriorating propagation conditions and shadowing effects.Matched Filter (MF): A technique based on correlation of the received signal with a known pattern. This is effective in the presence of additive white Gaussian noise (AWGN), but requires the signal type to be known [27,28]. The scope of MF use is limited to situations where we have full knowledge of the characteristic signal parameters, e.g., friendly force transmissions or known protocols.Spectrum monitoring based on covariance matrices: More resistant to the occurrence of interference and noise in the radio channel is the method of using the statistical properties of the covariance matrix of the received signals. In the case of the presence of a useful signal, additional components appear in the matrices, which allow the system to distinguish the radio signal matrix consisting only of noise from the matrix containing the signal [29,30].

Implementing the above-mentioned detection techniques on mobile platforms, e.g., UAVs, enables increasing the range of operation, introducing dynamic modifications to the analyzed area, and coordinating coverage of large areas.

### 2.2. Integration of Software-Defined Radio and Data Fusion Methods

There is a noticeable change required in the concept of using UAVs in spectrum monitoring, emitter location, or unauthorized access detection. Classic systems were usually equipped with a single sensor or effector, which allowed for monitoring, detecting emitters, or building situational awareness. This approach is easy to implement, but it has three basic limitations: limited spatial and temporal coverage (does not allow monitoring of large areas), susceptibility to noise or interference (data from a single sensor is easier to distort), and high rates of false alarm probability in complex operational environments. The above solution is opposed to cooperation between a group of sensors, which allows tasks to be performed by many platforms and data to be shared from sensors with a data fusion center, thus increasing the reliability of monitoring, enabling the localization of signal sources [31]. Multi-sensor systems also enable dynamic task reallocation, which allows for automatic adjustment of their position based on information obtained from the analyzed environment. Work [32] confirms that cooperation between sensors leads to increased resistance to interference, reduced erroneous results, and better allocation of available resources.

The effectiveness of cooperative detection depends not only on the implemented sensing method but also, to a large extent, on the data fusion algorithm (see Figure 2), which can be broadly divided into hard and soft.

In the case of HDF, each sensor sends a binary decision (0 or 1), defining the monitoring result. The data fusion center can apply one of the selected rules to make the final decision [33,34]:OR rule: The signal is detected if any of the sensors detects it;AND rule: The signal is detected only if all sensors make the same decision;K-out-of-N rule (majority voting): The signal is detected if K or more sensors report detection.

Despite its simple implementation and low computational requirements, hard data fusion is sensitive to individual sensor errors and cannot model uncertainty in the results. In soft data fusion, sensors send values of analyzed features (e.g., energy level, detection probability) to a data fusion center, which makes a global decision using weighted averages, probabilistic models, or belief networks. This increases performance, especially under uncertainty, but requires significantly higher transmission bandwidth and more computational resources [35].

An approach that is gaining popularity is data fusion based on DST. DST is an evidence-based method that allows the fusion of uncertain, inaccurate, or incomplete data. It introduces belief, probability, and weight functions, enabling the modeling of ignorance and partial truth. DST does not require complete a priori knowledge and can respond to conflicting information from multiple sensors. The above advantages are why DST is ideally suited for implementation in groups of UAVs operating in non-ideal conditions, similar to the requirements of the current battlefield. Technological developments also positively impact the possibilities of cooperative spectrum monitoring. The miniaturization of SDR components has enabled the installation of sensors on small UAVs. Platforms such as the Universal Software Radio Peripheral (USRP) B200 mini, HackRF, and RTL-SDR (i.e., Realtek RTL2832U chipset-based SDR) offer great possibilities at a relatively low price and small dimensions. Integration with UAVs allows for reconfiguring monitoring parameters in real time, collecting IQ (i.e., in-phase (I) and quadrature (Q) components) samples for processing and analysis of received signals, and operating in various bands (e.g., very high frequency (VHF)/ultra-high frequency (UHF), global navigation satellite system (GNSS), Long-Term Evolution (LTE), Wi-Fi). Thanks to the sensor architecture based on SDR, the system has the possibility of a dynamic and adaptive approach to the implementation of monitoring and electronic warfare (EW) tasks.

### 2.3. System Architecture

To contribute to developing cooperative spectrum monitoring systems in complex electromagnetic environments, this paper proposes a simulation-based test architecture developed in the MATLAB programming environment. The proposed architecture emulates a group of UAVs equipped with SDRs that act as spatially distributed radio sensors. The system’s main objective is to evaluate the detection performance of radio emitters using spectrum monitoring algorithms and DST-based data fusion, which are capable of modeling uncertainty and combining incomplete information from multiple UAVs. In the proposed architecture, each UAV is modeled as a mobile node equipped with an SDR module USRP B200 mini [36] and a microcomputer Raspberry Pi 4 [37], whose task is to manage the SDR operation and preprocess the received signals. The USRP B200 mini offers a small footprint, wide operating range, and sufficient bandwidth for spectrum monitoring, while maintaining low weight—an important factor when integrating sensors with UAVs. The Raspberry Pi 4 microcomputer was chosen for its sufficient processing power, small size, and low power consumption. With its 1.5 GHz quad-core processor, the Raspberry Pi 4 is capable of performing operations such as energy detection and preprocessing within the required 100 ms reporting interval. For analysis of a 20 MHz bandwidth, we need to process 2,000,000 samples in a 100 ms cycle. Due to the complex sample format and float32 data storage, we can assume that with a continuous data stream, we will achieve a throughput of 16 MB per cycle—160 MB/s. Using Universal Serial Bus (USB) 3.0 allows for processing up to approximately 400 MB/s. The energy detector performs two multiplications and two additions for each sample. With 2 million samples, the total number of operations is 8 million. The Raspberry Pi 4 can perform approximately 1.5–2 billion operations per second on a single core. With 8 million operations and a 1.5 GHz processor, the duration is approximately 8 ms. Even with the additional headroom, a larger number of operations, 100 ms is a sufficient reporting period. In the MATLAB environment, the behavior of each UAV is represented by simulation objects that

Sample the generated radio frequency (RF) data;Estimate the energy of the received signal samples in the analyzed frequency bands;Use the defined threshold values to determine the occupancy of the radio channel;Calculate the mass and belief functions, in the case of the implementation of data fusion based on DST.

The simulation also includes the representation of mobility and radio link quality degradation to reflect the real operating environment. The proposed system implements different scenarios, which allow for comparing the results of the DST approach.

Although the architecture is simulated, it is intended to reflect real-world conditions. Each UAV is assumed to transmit data from its sensors over a Wi-Fi-based channel to a ground-based data fusion center. MATLAB models this exchange using data structures representing asynchronous packet-based communication. Latency and packet loss models are included to assess the system’s robustness under degraded communication conditions. The data fusion center in the simulation is responsible for receiving data from all sensors, applying the selected data fusion algorithm, making the final decision on the presence of emitters, and generating the system performance characteristics. The fusion center is implemented in MATLAB. The simulation scenario assumes the deployment of various UAVs, each of which monitors a given frequency band in which the emitter may be located. The emitter signal is modeled as a narrowband transmission with time-varying SNR depending on the UAVs’ location, representing propagation and interference effects. The UAVs perform tasks according to a defined monitoring schedule and report results every 100 ms. The simulation includes the following: the movement of emitters, time-varying emitter activity, varying interference levels, and the mobility of the UAVs (UAVs change their height and position). The simulation setup consists of two mobile TXs and one static TX. The TXs move in opposite directions, moving away from each other and away from the static emitter. The starting location of the TX is randomly generated in each iteration, but it is approximately 1500 m in the *x*-axis from the point (0, 0) and 500 m in the *y*-axis. The sensors always move towards the TXs, despite changes in the TXs’ positions. Additionally, the sensor group moves in one of three proposed formations: scattered, V-shaped, and linear. Regarding the additional interference sources implemented, the main distinction is the simulation of various terrain and building models: urban, suburban, and dense urban. The radio noise levels were implemented in accordance with the International Telecommunication Union (ITU) recommendations—ITU-R P.372 [38]. In each iteration, the data fusion center collects data from the sensors and decides on the presence of the emitter. Then, the detection performance of the DST fusion method is compared. The simulation results, presented in the next section, show that such a system increases detection accuracy, reduces false alarms, and improves situational awareness in scenarios reflecting the real electromagnetic spectrum conditions during a realistic scenario for using telecommunications devices.

### 2.4. Implemented Algorithms

In the proposed system, radio emitters are detected using a network of sensors mounted on UAVs, acting as EDs. Considering the real-time and resource-constrained nature of UAV operations, energy detection offers a practical balance between simplicity of implementation and high efficiency. After data preprocessing, the results are transmitted to a ground-based data fusion center, where the DST fusion method is used.

#### 2.4.1. Energy Detector

The ED is one of the most commonly used implementations of spectrum monitoring due to its simplicity, generality, and low computational requirements. In the proposed solution, each UAV uses its sensor to receive IQ samples of the signal from the analyzed frequency band and calculates the test statistic according to the following formula:
(1)$$T=\sum_{k=-\infty }^{\infty }{\left|y\left(k\right)\right|}^{2},$$
where $y\left(k\right)$ is the *k*-th received complex sample of the signal, composed of the transmitted signal component $s\left(k\right)$ and additive noise $n\left(k\right)$. The detector compares the test statistic with a predefined threshold, calculated based on
(2)$$z=\frac{1}{2N\sigma^{2}}\sum_{k=1}^{N}{\left|y\left(k\right)\right|}^{2},$$
where $\sigma^{2}$ is the noise variance and *N* means the number of samples. The proposed solution enables rapid decision-making, which is essential in UAV operations.

The main advantages of EDs in this context include

Low computational requirements;Wide application, independent of the received signals;Fast detection, enabling frequent updates and flexibility.

However, EDs are also sensitive to noise and the impact of multipath effects, justifying the need for cooperative monitoring and the implementation of data fusion algorithms.

#### 2.4.2. Data Fusion

Given the spatial distribution of the sensor network on the UAV, local decision fusion is essential to improve overall detection reliability. In this research, we focused on comparing the performance of data fusion based on the DST method.

DST is used for modeling and reasoning about uncertainty, making it more useful in dynamic operational scenarios (e.g., intermittent communication between UAVs, measurements affected by noise). In spectrum monitoring, each UAV acts as an individual sensor, making an independent decision and assigning degrees of confidence to possible hypotheses regarding the state of the monitored frequency band. Based on the original publication of the authors of the method [39,40], which described the theoretical basis of the method, let us define a frame of discernment as
(3)$$\Theta =\{H_{0},H_{1}\},$$
and the corresponding power set includes
(4)$$2^{\Theta }=\{Ø,\{H_{0}\},\{H_{1}\},\{H_{0},H_{1}\}\},$$
where Ø denotes the empty set, $H_{0}$ and $H_{1}$ are the hypotheses that the channel is free (no signal) and occupied (signal present), respectively, and $\Theta =\{H_{0},H_{1}\}$ represents the uncertainty mass. For this frame of discernment, it indicates that the sensor has evidence that the true state lies somewhere within the frame, but lacks sufficient information to support either hypothesis individually. It may occur when the SNR is too low to distinguish the presence of a signal, there is severe interference or shadowing, or received signals are corrupted or incomplete.

Each UAV calculates a basic probability assignment (BPA) based on its local decision:
(5)$$m:2^{\Theta }\to [0,1],\sum_{A\subseteq \Theta }m(A)=1,m(Ø)=0,$$
where m is the mass function and A is a subset of the set of hypotheses.

The BPA assigns probability masses to individual hypotheses $m(\{H_{0}\}),m(\{H_{1}\})$ and the whole frame m(Θ) representing uncertainty.

To fuse information from multiple sensors, Dempster’s rule of combination is applied to combine multiple BPAs, $m_{1},m_{2}\ldots m_{n}$, into a new BPA m according to the formula below:
(6)$$m(A)=\frac{1}{1-K}\sum_{B\cap C=A}m_{1}(B)\cdot m_{2}(C),$$
where m(A), m(B) and m(C) are combined mass functions assigned to set *B* by the first sensor, and assigned to set *C* by the second sensor, respectively, and K is the conflict coefficient, which measures the total weight assigned to conflicting information.

The conflict coefficient K is defined as(7)$$K=\sum_{B\cap C=\Theta }m_{1}(B)\cdot m_{2}(C),$$

The principle of fusion can be used to combine information from more than two sources.

When the BPA is combined, two key functions are defined:Belief function (Bel)—The sum of all the masses supporting a given hypothesis:(8)$$Bel(A)=\sum_{B\subseteq A}m(B),$$

Plausibility function (Pl)—The sum of all the masses that do not contradict the hypothesis:


(9)
$$Pl(A)=\sum_{B\subseteq A\neq Ø}m(B),$$


A final decision can be made by comparing the belief in or plausibility of a hypothesis $H_{1}$ with a chosen threshold λ:(10)$$Bel(H_{1})>\lambda \Rightarrow \mathrm{Channel}\mathrm{is}\mathrm{occupied},$$

We used an adaptive decision threshold to determine the threshold value. This is a low-complexity, high-performance, and highly flexible solution. The performance and threshold depend primarily on two factors: the expected false alarm probability and noise power. It can be defined as
(11)$$\lambda =\delta^{2}\left[\sqrt{2\cdot N}\cdot Q^{-1}(P_{fa})+N\right]$$
where $\delta^{2}$ represents the variance of noise, *N* represents the number of samples, and *Q* represents the right tail function of the standard distribution.

DST-based fusion is more robust to uncertainty and contradictory evidence than hard fusion methods because it allows for partial belief and ignorance. This makes it particularly effective for monitoring the electromagnetic environment, where sensors may operate under high interference or a lack of visibility.

## 3. Results

In this study, we analyze the effectiveness of proposed fusion methods for electromagnetic spectrum monitoring using UAVs. The simulation is conducted in MATLAB, implementing a scenario in which multiple UAVs are deployed to detect the presence of a signal TX in a given area.

Three different types of scenarios are considered for this simulation:Environment type (urban, suburban, dense)—Each UAV is moving through a different type of terrain. Propagation conditions were simulated using MATLAB tools and considered the influence of various factors depending on the selected scenario. The urban scenario represents a city with multi-story buildings (3–5 stories) and considers signal reflections from buildings and the presence of other signal sources in the analyzed area. The suburban scenario considers an open space with single low-rise buildings, few other emission sources, and fewer obstructions. Dense urban development presents the most challenging conditions for spectrum monitoring. This scenario features narrow streets, tall buildings that reflect signals, and high levels of mutual interference with other sources.Formation of UAV swarm (linear, V-shaped, scattered)—In a linear formation, the UAVs are positioned 75 m apart and fly in the same direction, maintaining their distance. Twenty seconds before the simulation ends, they approach each other within 5 m. This configuration allows uniform coverage of a wide area (300 m span). This configuration allows for high variability in the obtained results while maintaining low measurement uncertainty. In a linear formation, the UAVs are positioned 75 m apart and fly in the same direction, maintaining their distance. Five seconds before the simulation ends, they approach each other within 5 m. This configuration allows uniform coverage of a wide area (300 m span). This configuration allows for a high degree of variability in the obtained results while maintaining low measurement uncertainty. In the randomly dispersed scenario, the UAVs are located between 50 and 150 m apart, and the coverage varies for each scenario. Another factor is the variable flight altitude, which was set at 100 m in the previous scenarios but ranges from 50 to 150 m in this case.Number of UAVs (3, 5, 7, and 10)—The analyzed number of UAVs should improve detection efficiency by increasing the diversity of received signal samples, but it may also increase conflicts between individual sensors. The main goal of changing the UAV group size is to verify its impact on the implemented data fusion algorithm and determine whether there is a threshold beyond which further increasing the number of UAVs is no longer effective.

The scenario models varying environmental conditions and SNRs and evaluates detection performance using a comparison of the probability of detection ($P_{d}$) with the probability of false alarm (*P_fa_*) and metrics related to fusion method—belief (Bel), plausibility (Pl), uncertainty, and conflict.

### 3.1. Simulation Setup

The simulation considers a cooperative spectrum monitoring system comprising a variable number of UAVs operating over a three-dimensional area, specifically {3, 5, 7, 10}. Within this area, three TXs are located; two are on the move and emit radio waves at a known frequency. The UAVs autonomously fly over the area, collect data (i.e., IQ samples), and estimate energy levels depending on the chosen sensing strategy. All collected data is then transmitted to a data fusion center, which processes the information using DST fusion techniques to detect the TX’s presence, depending on the method and the number of UAVs used.

#### 3.1.1. Signal and Channel Model

The simulation scenario assumes the use of a private mobile radio (PMR) as a TX. In this case, differential phase shift keying (DPSK) modulation was used to simulate signal transmission. The TX operates at a frequency of 150 MHz with a bandwidth of 25 kHz and a sampling rate of 100 kHz. The transmission of individual packets takes approximately 10.24 ms. The assumed transmit power is 3.5 W. These assumptions allow for good propagation in urban areas with a range of several kilometers. Each TX emits a different, random bit sequence regarding the transmitted data. For each UAV *i*, the received signal is expressed as(12)$$y_{i}(t)=s(t)+n_{i}(t),$$
where s(t) is the transmitted DPSK signal, $n_{i}(t)$ is zero-mean Gaussian noise with variance $\delta^{2}$. The simulation used a pseudorandom binary sequence mapped to DPSK symbols, where amplitudes of −1 and 1 corresponded to binary 0 and 1. The signal was generated in 1000 iterations, and the obtained results were averaged over the number of loop iterations.

The signal propagation follows an Okumura–Hata model, where the equation gives the path loss (*PL*) in decibels [41]:(13)$$\begin{array}{ll} PL(dB) & =69.55+26.16\log_{10}(f)-13.82\log_{10}(h_{T})-a\log_{10}(h_{R})\\ & +(44.9-6.55\log_{10}(h_{T}))\cdot \log_{10}(d), \end{array}$$
where d is the distance between the emitters and the receiving UAVs in km, *f* is the carrier frequency of the transmitted signal, $h_{T}$ is the TX antenna height, and $h_{R}$ is the receiver (RX) antenna height. In the simulations performed, the height of the sensor placed on the UAVs was considered for *PL* calculations under various propagation conditions. The distance *d* was evaluated over the full operational range of the UAV formation (up to 4 km depending on scenario). This model reflects the signal attenuation with distance and frequency in different environments, serving as a basis for evaluating detection performance under different deployment configurations and fusion strategies. Depending on the chosen scenario, additional propagation losses arise from shadowing, interference, and reflections from buildings. An example scenario (linear formation) is presented in Figure 3.

The received signal samples were analyzed and used for ED implementation. Results were sent for further analysis at the data fusion center. An ED was implemented, which compared the obtained energy value with a predefined threshold.

#### 3.1.2. Fusion Methods

Dempster–Shafer fusion requires a more nuanced approach than hard-decision or soft-decision fusion. Each sensor provides a BPA based on its own observations. The BPA includes the values $m(H_{0})$, *m*(*H*_1_) and *m*(*U*), where *m*(*U*) represents the degree of uncertainty. Then, using Dempster’s combination rule, the fusion center combines BPA data from multiple sensors. The combined BPA score is calculated as the normalized sum of the products of the individual tasks, excluding contradictory evidence.

Each sensor receives data regarding detection probability, binary decision, or IQ samples during the simulation loop. Each sensor’s observation is used to construct a mass function for three components: {signal}, {noise}, and {uncertainty}. The BPA values for each sensor are stored in a matrix. The local mass assignments are then computed in a loop. The sensor confirms the signal hypothesis if the test statistic is above the threshold. Otherwise, it confirms the noise hypothesis. In both cases, a constant fraction (10%) of the belief is assigned to the uncertainty. The BPA values are iteratively combined according to Dempster’s combination rule. Fusion begins with the first mass vector. This is then sequentially combined with the BPA values of the subsequent sensors. A new combined mass is then calculated. After combining all BPA sensors, a final decision is made. That is, a positive detection is made if the belief mass for the signal hypothesis exceeds the belief mass for the noise hypothesis. The uncertainty mass is not used for decision-making but plays a key role during fusion. The entire process is repeated for 1000 iterations and with different numbers of UAVs.

### 3.2. Simulation Results

Several performance metrics were considered to evaluate the proposed data fusion method’s effectiveness. Key metrics were belief, plausibility, probability of detection, and probability of false alarm. These metrics enabled the comparison of detection performance under different operating conditions.

The impact of environmental conditions, formation, and the number of UAVs participating in spectrum monitoring was analyzed. Simulations were conducted according to the planned scenario (see Figure 3), and in the first simulation, we wanted to examine the impact of changing parameters on the probability of detection. To conduct such a task, we chose to average the results from the whole simulation—60 s. System performance was evaluated in scenarios involving three, five, seven, and ten UAVs to observe how increasing the number of cooperating sensors impacts detection accuracy. Different formations (linear, V-shaped, and scattered) and propagation environments (urban, suburban, and dense) were considered. The results are presented in Figure 4. In Figure 4, we used the following description of the analyzed scenario cases:3UAV-lin-urb means three UAVs flying in linear formation in the urban scenario,5UAV-lin-urb means five UAVs flying in linear formation in the urban scenario,7UAV-lin-urb means seven UAVs flying in linear formation in the urban scenario,10UAV-lin-urb means ten UAVs flying in linear formation in the urban scenario,5UAV-vshp-urb means five UAVs flying in V-shape formation in the urban scenario,5UAV-sca-urb means five UAVs flying in scattered formation in the urban scenario,5UAV-lin-sub means five UAVs flying in linear formation in the suburban scenario,5UAV-lin-dense means five UAVs flying in linear formation in the dense scenario.

This abbreviation is used for the rest of the paper.

Figure 4 compares the performance of eight data fusion scenarios for three working emitters. The evaluation metric was the probability of detection $P_{d}$. Each method was analyzed for different scenarios: sensors mounted on three, five, seven, and ten UAVs; in linear, V-shaped, and scattered formations; and in urban, suburban, or dense environments. This allowed us to assess the impact of increasing the number of sensors and changes in propagation effects on detection performance. On the graph, emitter detection performance increases with the increase in the number of sensors used. In a linear configuration in an urban environment with three sensors, the detection probability is approximately 0.7 per TX. It increases with a higher number of UAVs—0.83 for five UAVs, 0.89 for seven UAVs, and 0.93 for ten UAVs. While a growing number of sensors increases the probability value, the next four bars show how different formations and environmental effects affect detection efficiency. All four were set up with five sensors, and we obtained the highest probability from the one operating on suburban terrain due to the better propagation conditions and lower PL and reflection factor. The detection probability achieved 0.87, slightly higher than the V-shaped formation in urban environments. From these results, we can conclude that the V-shape is more effective, because working in worse conditions achieves comparable results, *P_d_* from 0.83 up to 0.86. The two lowest *P_d_* results come from the scattered formation and dense environment—0.77 and 0.78, respectively. It emphasizes that propagation conditions play a crucial role in energy detection algorithms. Using a different formation for the UAV group also matters and impacts algorithm efficiency. The highest detection probability was achieved when the V-shaped formation was chosen. For the same scenario (five UAVs, urban terrain), the probability was equal to 0.83 for linear, 0.85 for V-shaped, and 0.78 for scattered.

The next object of analysis was a comparison of different scenarios and their influence on the probability of a false alarm. The setup was the same as in the previous test, with a different formation, propagation environment, and number of UAVs. The results are presented in Figure 5.

The probability of false alarm represents the chances of obtaining information about a busy channel in the absence of a transmitted signal. The higher the value, the worse the system performance. In our research, the results obtained are consistent with the previous conclusions. With greater UAVs, the probability of false alarm decreases—from 0.14 for three UAVs down to 0.07 for ten UAVs. Working in dense environments causes worse propagation conditions and a higher probability of false alarm. The results for the linear formation with five UAVs in dense, suburban, and urban terrain are 0.08, 0.09, and 0.12, respectively. The chosen formation can also negatively influence detection algorithms—a change from a V-shaped formation to a scattered one can increase the probability of false alarm from 0.09 up to 0.13.

To better visualize the impact of each parameter, we have prepared a comparison of the impact of different formations (Figure 6) using five UAVs in urban or different environments (Figure 7) and in a linear configuration on the detection and false alarm probabilities.

Figure 6 shows the monitoring performance—probability of detection and probability of false alarm for different UAV formations in the case of five UAVs and urban environments. It is clear that the scattered formation is the worst one. It has a probability of detection of 0.08 lower and a probability of false alarm 0.03 higher than the other approach. This may be due to the formation’s random nature and the sensors’ interactions. Sensors placed in close proximity and disorganized may report similar results, whereas sensors farther apart may yield different values, leading to conflicting decision-making.

Figure 7 highlights the influence of different environments on the probabilities of detection and false alarm. Due to a lower PL, signal reflection, fewer obstacles, and multipaths, sensors working in suburban terrain can achieve a higher probability of detection (0.86) and a lower probability of false alarm (0.08) for five UAVs in a linear formation. In dense or urban terrain, these parameters are worse, and the dense urban scenario has the lowest probability of detection (0.77) and the highest probability of false alarm (0.13).

In the case of analyzing the probability of detection and the probability of false alarm, the last object of analysis was an RX operating characteristic for different scenarios (see Figure 8).

The graph illustrates the detection and false alarm probabilities for different scenarios. In this type of visualization, ten UAVs in a linear formation and an urban scenario outperformed other approaches. It is clear that having more sensors results in a higher probability of detection and a lower probability of false alarm. What is worth noting is the comparison of the results obtained for the same number of sensors—five UAVs. In this case, we can clearly see the strong influence of the chosen formation and environment type, as in the previous analysis. A linear formation in a suburban environment achieves a 0.86 probability of detection and a 0.08 probability of false alarm. Even a change from suburban to urban conditions affects the obtained results—for the V-shaped formation, the probability of detection decreases to 0.84, and for the linear formation to 0.82, for the five-UAV case. As the number of sensors decreases, the detection probability value also decreases, and the probability of false alarm increases—for three UAVs, it reaches the value 0.7.

To confirm the above, the parameters of DST data fusion were also analyzed for an identical situation—using three, five, seven, or ten UAVs in different scenarios and configurations. The results for the belief function are presented in Figure 9.

The belief function for each previously described scenario varies depending on the number of sensors, the environment, the formation, and the TX from which the signals are received. TX1 and TX2 were moving, while TX3 remained stationary throughout the simulation. As a result, we obtained higher belief function values for the stationary emitter. A UAV swarm moves toward a stationary target when movable emitters move away in different directions. Analyzing the obtained values, the same trend is observed as in the case of probability of detection and false alarm—an increase in the number of sensors yields higher belief function values. The influence of the chosen environment and formation also matters—it is clear that five UAVs in suburban terrain and flying in linear formation give better results than the same number of sensors in different environments and formations. To be more precise, the best case was five UAVs in a linear formation and working in suburban terrain. The belief function reached a value from 0.9 to 0.97, depending on the emitter type. The same number of sensors, but in an urban area and flying in scattered formation, had a lower belief function value—0.51 for the first moving emitter, 0.3 for the second one, and 0.67 for the stationary emitter. Another scenario yielded similar results, with a visibly higher belief function value for stationary emitters. What is worth noting is the minimal decrease in the belief function value when increasing from seven UAVs to ten UAVs—from 0.68 to 0.67. A larger number of sensors may lead to conflicts between individual elements, which lowers the final result.

We chose the plausibility function to be the next analyzed parameter for the same scenario as before. This allowed us to confirm the above theses for the DST fusion. The results are presented in Figure 10.

This time, a different tendency can be observed—as the number of sensors increases, the value of the plausibility function decreases. For example, for an urban scenario with three UAVs in linear formation, the value is 0.71 for a stationary emitter, but for ten UAVs, it is only 0.67. The plausibility function value for seven and ten UAVs is almost equal to the value of the belief function for the same scenario. A decrease in plausibility can occur when more sensors provide more or more consistent evidence supporting the opposite hypothesis $H_{0}$. In such a situation, the function Bel increases, which directly decreases the value of Pl. Our observations reflect exactly this mechanism: when many sensors provide convergent evidence pointing to a free channel, the confidence in *H*_0_ increases, and the upper bound of the supporting hypothesis *H*_1_ naturally decreases. At the same time, in scenarios in which sensors provide more results confirming the presence of a signal, Pl remains higher than Bel, as expected, creating a valid uncertainty interval. This proves that the plausibility function behaves as theoretically expected—its change results from the concentration of evidence in competing hypotheses and not solely from the number of sensors. In the case of five UAVs in a linear formation and a suburban environment, which produces the highest values of the belief function, we can observe the highest results, but with only a slight increase from 0.91 to 0.99. In the case of a smaller number of sensors, the tendency is consistent with the assumptions—the values of the plausibility function are greater than those of the belief function and constitute the upper range of probability, whereas belief is its lower range.

Another important parameter from the DST fusion perspective is the uncertainty value. It defines the level of knowledge gap between certainty and acceptability. The values of this function are influenced by the uncertainty resulting from missing information or inconsistent data. We wanted to confirm the correctness of the data received in the plausibility function. The results for the uncertainty function are presented in Figure 11.

The obtained values are low, below 0.1 in each scenario, and the highest is for three UAVs in linear formation and urban terrain, from 0.8 to 0.11. The higher the value of the uncertainty function, the greater the difference between the belief and plausibility functions shown in the two previous graphs. It is clear that, for seven or ten UAVs in the same scenario, the uncertainty values are very low, below 0.01, which explains why the plausibility function values are almost equal to the belief function values. The decrease in uncertainty is due to the increased number of UAVs, leading to greater evidence consistency during the fusion process. Scenarios for five UAVs with the V-shaped and scattered formations are characterized by higher uncertainty than those for the linear formation, despite the identical number of drones. The influence of terrain may result from differences in spatial coverage and observation angles—irregular formations cause greater measurement discrepancies, translating into greater evidence conflict.

The final analysis was based on conflict values. DST fusion defines the degree of contradiction between information sources. The higher the conflict, the more the data from different sensors contradict each other. A high K means the fusion is less credible because the sources disagree on the same hypothesis. The results are presented in Figure 12.

In a linear formation and urban terrain, the conflict value is approximately 0.2 for three UAVs, while for ten UAVs, it reaches a maximum of ~0.47. This indicates that increasing the number of sensors in a fixed-geometry system leads to greater discrepancies in the evidence masses, i.e., increased information conflict between sensors. For scenarios with five UAVs, the impact of deployment and flight geometry on the conflict value is noticeable: the V-shaped and scattered formations exhibit moderate conflict levels (~0.35), suggesting that diverse observation angles and greater spatial coverage in nonlinear systems (V-shape, scattered) may partially compensate for the mutual measurement inconsistencies and reduce conflict. Comparison of the urban and suburban scenarios for five UAVs reveals a significant decrease in conflict in the suburban environment—in the urban environment, *K* ≈ 0.35, and in the suburban environment, *K* < 0.2. This means that in a less noisy radio environment, sensors record more consistent data, thereby reducing conflict levels in the DST fusion process.

To more accurately illustrate the trends and characteristics of individual solutions, the obtained results are also presented in tabular form. Table 1 presents how different environments affect energy detection algorithms and data fusion metrics.

Data analysis shows that with increasing building density, both the belief and plausibility functions decrease, indicating a decreasing degree of certainty and consistency in the information provided by the sensors. At the same time, uncertainty and conflict levels increase, reflecting the growing difficulty of data fusion in environments with a stronger multipath and greater signal attenuation.

A similar trend is evident in the detection parameters: the detection probability decreases from 0.85 (suburban) to 0.77 (dense), while the false alarm probability increases from 0.08 to 0.12. This indicates that the detection system loses its effectiveness in densely built-up environments, confirming the negative impact of the urban environment on electromagnetic wave propagation, and the decision-making process within the DST analysis of the data fusion metric leads to a similar conclusion—the values of the belief and plausibility functions systematically decrease with increasing building density, from 0.81/0.98 (suburban) to 0.72/0.96 (dense), respectively. This indicates that data fusion provides less certain and less reliable conclusions in environments with more complex signal structures. A decrease in the belief value indicates a limited amount of unambiguous evidence confirming the detection, while a lower plausibility indicates that an increasing proportion of the belief mass is attributed to contradictory or uncertain hypotheses.

The next table (Table 2) provides insight into how different formations of UAVs affect overall performance and values.

The belief and plausibility function values are highest for the V-shaped formation (0.80 and 0.98), indicating that this arrangement provides information consistency and the highest confidence level in the fusion results. For the linear formation, these values are slightly lower (0.78 and 0.97), while for the scattered configuration, they reach a minimum (0.72 and 0.96), indicating a deterioration in information quality.

The uncertainty and conflict parameters confirm this relationship—the lowest levels of uncertainty (0.17) and conflict (0.09) are observed for the V-shape, suggesting that this UAV deployment geometry minimizes discrepancies between sensors and promotes effective data fusion. On the other hand, the scattered configuration is characterized by the highest values of these metrics (0.24 and 0.14), indicating greater differences in observations and difficulty in combining information into a coherent result. These trends are consistent with the values of classical detection metrics: detection probability is highest for the V-shaped configuration (0.85) and lowest for the scattered configuration (0.76). At the same time, the false alarm probability is lowest for the V-shaped configuration (0.09), confirming the optimal compromise between sensitivity and selectivity of the system. The results indicate that the UAV deployment geometry significantly impacts the quality of DST fusion and detection efficiency. The V-shaped configuration provides the best observation conditions and data consistency, leading to the highest belief and plausibility values, while simultaneously having the lowest levels of conflict and false alarms.

Table 3 provides information about the influence of the UAV number on overall performance.

A clear improvement in all key parameters is noticeable as the number of sensors increases. The belief function value increases from 0.64 (for three UAVs) to 0.89 (for ten UAVs), indicating increased decision confidence resulting from greater evidence confirming signal detection. At the same time, uncertainty decreases systematically from 0.29 to 0.09, indicating reduced uncertainty and greater clarity in fusion results. A similar trend is observed for conflict (Conflict), which decreases from 0.16 to 0.05, confirming that with more UAVs, data from individual platforms is more consistent and less contradictory. Parallel to the improvement in DST metrics, the detection probability increases—from 0.70 for three UAVs to 0.93 for ten UAVs—demonstrating a significant increase in system effectiveness. At the same time, the false alarm probability decreases from 0.142 to 0.065, confirming that increasing the number of platforms improves the system’s selectivity and resilience to false detections.

The obtained results clearly show that a larger number of UAVs in the system leads to more reliable and stable data fusion per the DST, increasing detection accuracy and reliability. Increasing the number of sensors allows for a more effective reduction in uncertainty and information conflict, translating into better decision-making parameters for the entire system.

## 4. Conclusions

This paper evaluated cooperative spectrum-monitoring systems based on UAV swarms equipped with SDR sensors, particularly with data fusion methods for detecting radio emitters in complex electromagnetic environments. The research demonstrated that integrating multiple UAVs with a DST data fusion algorithm significantly improves detection efficiency compared to traditional single-sensor approaches.

The simulation results provided several conclusions about the performance of UAV-based spectrum monitoring systems:Increasing the number of UAVs from three to ten positively affects detection performance. The probability of detection increased from 0.70 for three UAVs to 0.93 for ten UAVs in urban environments with linear formation, while the probability of false alarm decreased from 0.14 to 0.07. However, the analysis of DST metrics revealed diminishing returns beyond seven UAVs, with the belief function showing minimal improvement when expanding to ten UAVs. For the belief value, the increase between three UAVs (0.62) and seven UAVs (0.68) is only about 9.5%, and for ten UAVs, the value does not increase. This suggests the existence of an optimal threshold at which additional sensors may introduce conflicts among individual sensors, offsetting the benefits of increased spatial coverage.Propagation conditions emerged as an important factor affecting system performance. In suburban environments with reduced multipath effects and signal attenuation, the system achieved detection probabilities of 0.86 with false alarm rates as low as 0.08. In dense urban scenarios, where significant signal reflections and interference occurred, the probability of detection decreased to 0.77, and the probability of false alarm decreased to 0.13. The DST metrics confirmed these trends—belief function values decreased from 0.81 in suburban to 0.72 in dense environments, while conflict coefficients increased from 0.08 to 0.14, indicating greater inconsistency between sensor observations in challenging propagation conditions.UAV deployment formation demonstrated a substantial impact on monitoring effectiveness. The V-shaped formation outperformed linear and scattered configurations, achieving 0.85 detection probability with only 0.09 false alarm probability for five UAVs in urban terrain. This configuration provided optimal observation angles and spatial diversity, resulting in the lowest uncertainty (0.17) and conflict (0.09) values. Despite offering the widest spatial coverage, the scattered formation exhibited the poorest performance (0.76 detection probability, 0.13 false alarm probability) due to irregular sensor spacing that generated contradictory measurements and elevated conflict levels (0.14).The DST-based approach demonstrated superior robustness under degraded signal conditions. The framework’s ability to model uncertainty through belief and plausibility functions provided explicit confidence bounds for detection decisions. Analysis revealed that as the number of sensors increased and measurements became more consistent, the gap between belief and plausibility functions narrowed—uncertainty values decreased from 0.28 for three UAVs to below 0.01 for ten UAVs in stable scenarios. The conflict value also decreased even under strong interference conditions in the urban scenario—for three UAVs, it was 0.16, and increasing to ten UAVs resulted in a value of 0.05—a 68.8% decrease. This convergence indicates high confidence in fusion outcomes when sufficient consistent evidence is available.The conflict value proved to be a valuable indicator of system reliability. Higher conflict values (*K* > 0.3) typically corresponded to scenarios in which environmental conditions or formation produced incompatible sensor observations. In suburban environments with five UAVs in linear formation, conflict remained below 0.2, whereas urban scenarios with scattered formations exhibited conflict values approaching 0.35–0.40. This metric enables real-time assessment of fusion reliability and can trigger adaptive responses when evidence inconsistency threatens decision quality.

The findings carry significant implications for the design and deployment of operational spectrum monitoring systems. System designers must balance the benefits of additional sensors against increased complexity and potential conflicts. The research indicates that five–seven UAVs represent an optimal range for most scenarios, providing substantial performance improvements over minimal configurations while avoiding the diminishing returns and elevated conflict levels observed with larger swarms. The strong influence of formation geometry suggests that adaptive deployment algorithms could dynamically adjust UAV positions based on current environmental conditions and detection requirements. V-shaped formations offer advantages in challenging environments, while linear formations may suffice in open terrain with good propagation conditions.

System performance prediction requires accurate characterization of the electromagnetic environment. Pre-mission analysis of expected propagation conditions should inform deployment parameters, sensor thresholds, and the configuration of the fusion algorithm to optimize detection capabilities. The DST metrics—particularly uncertainty and conflict coefficients—provide valuable real-time system reliability indicators. These measures can support operator decision-making by flagging situations where fusion results may be unreliable due to contradictory evidence or insufficient information.

This research helps understand cooperative UAV-based spectrum monitoring, but several limitations have been found. The simulation cannot fully capture the complexity of real-world operational environments. The energy detection method represents only one approach to spectrum sensing. Future research should investigate the performance of DST fusion with different sensing techniques—matched filtering, covariance-based detection, or cyclo-stationary feature detection. It will allow us to determine the optimal combinations for various operational scenarios. Developing real-time DST fusion algorithms on actual UAV platforms equipped with SDR modules will validate simulation predictions and reveal implementation challenges. Integrating DST with machine learning techniques and a deep reinforcement learning approach [42,43] could leverage the uncertainty modeling capabilities of DST while exploiting pattern recognition and adaptive learning offered by neural networks. Such hybrid approaches may enhance performance in highly dynamic or previously unencountered scenarios. Investigating adaptive positioning algorithms that optimize UAV deployment based on real-time DST metrics (particularly conflict and uncertainty) could enable autonomous adjustment to maximize monitoring effectiveness as environmental conditions evolve. Incorporating more detailed propagation models that account for weather effects, temporal variations, and three-dimensional terrain features will improve system accuracy in diverse operational contexts. Integrating electromagnetic spectrum data with other sensors (optical, thermal, acoustic) through DST fusion could provide situational awareness not only in the electromagnetic domain alone. Testing system robustness against countermeasures such as spoofing or jamming will be important for security applications.

This research demonstrates that cooperative UAV-based spectrum-monitoring systems employing DST fusion significantly enhance electromagnetic situational awareness capabilities. The framework’s ability to model uncertainty, handle incomplete information, and integrate potentially contradictory evidence from distributed sensors addresses fundamental limitations of conventional approaches.

The simulation results validate that properly configured UAV swarms can achieve detection probabilities exceeding 0.90 with false alarm rates below 0.08 under favorable conditions, while maintaining acceptable performance (>0.75 detection probability) even in challenging dense urban environments. Explicitly modeling uncertainty and conflict through DST metrics provides valuable indicators supporting decision-making.

As wireless communications continue to expand and the electromagnetic spectrum resources become limited, the need for spectrum monitoring capabilities will only intensify. UAV-based systems offer unique mobility and rapid deployment advantages. The integration of advanced data fusion algorithms such as DST enables these systems to operate effectively despite the challenges of real-world scenarios.

The next steps in our research involve transitioning to operational deployment, validating theoretical results through field tests, and continuing to improve algorithms based on practical experience. Although the presented results demonstrate the potential for collaborative monitoring using UAVs, several challenges can be expected in real-world hardware implementations. First, time and frequency synchronization between individual sensors, as low-cost SDR platforms experience clock drift and variable latency during information exchange. Second, bandwidth limitations of communication channels, especially when using small SDR platforms. Third, processing power limitations of the Raspberry Pi or other single-board computers, which may result in limited bandwidth or reporting frequency, necessitate the implementation of additional reporting and data-compression strategies. Considering all of these factors will enable an effective transition from simulation studies to practical testing. With ongoing advances in UAV technology, sensor miniaturization, and computational capabilities, cooperative spectrum monitoring systems are becoming essential tools for electromagnetic spectrum awareness and cognitive radio applications in civilian and military domains.

## Figures and Tables

**Figure 1 sensors-25-07298-f001:**
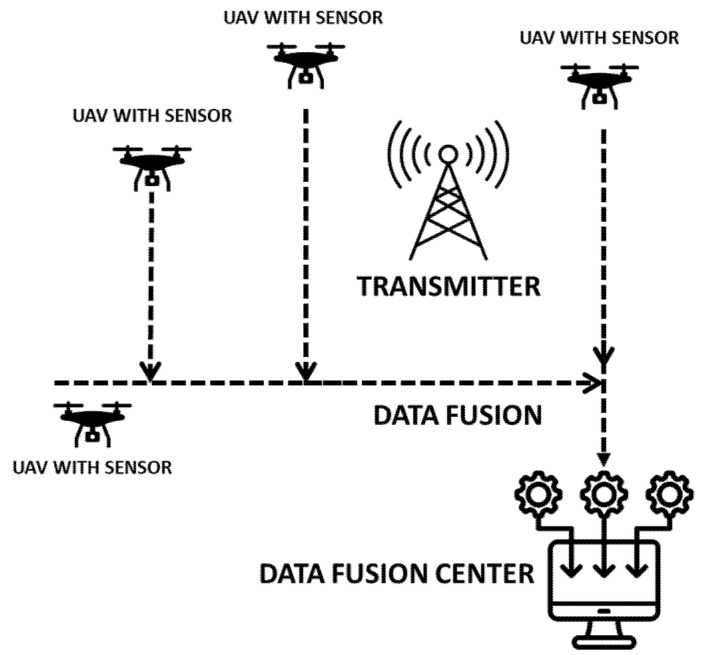
Concept of utilizing a UAV group for spectrum monitoring.

**Figure 2 sensors-25-07298-f002:**
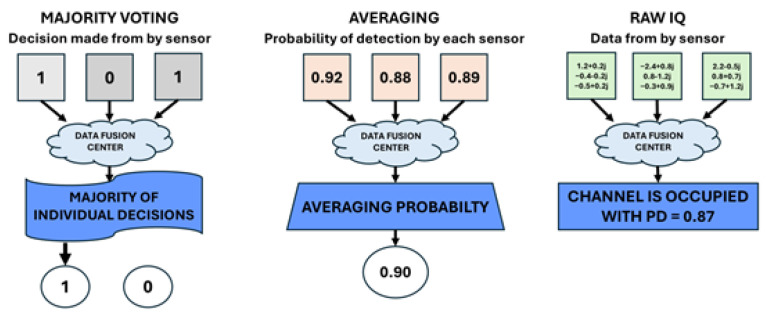
Different types of data fusion methods.

**Figure 3 sensors-25-07298-f003:**
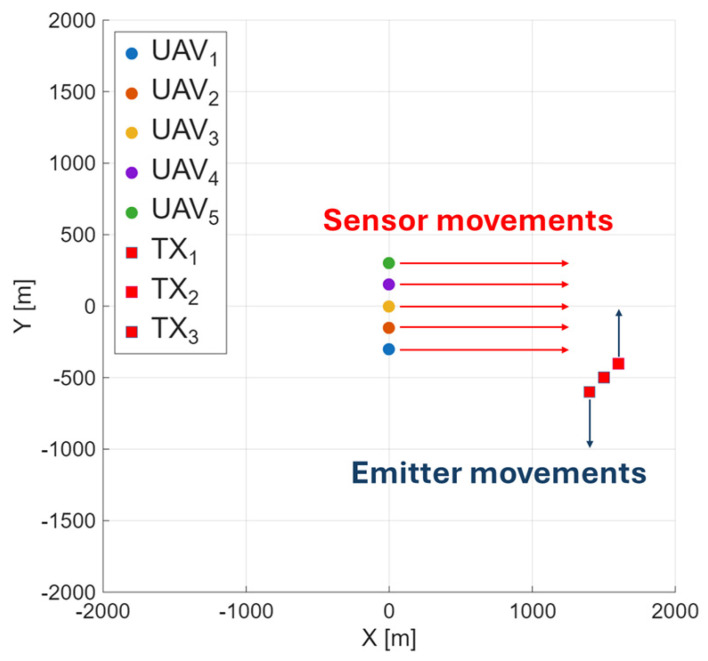
Received power estimates shown as a heatmap.

**Figure 4 sensors-25-07298-f004:**
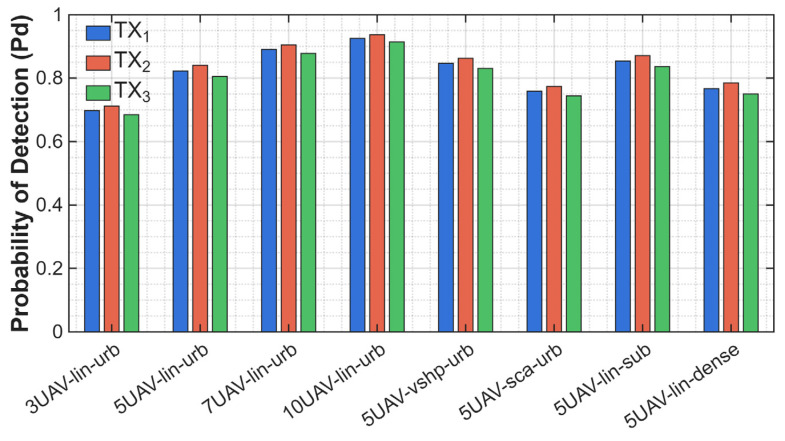
Comparison of scenarios and the influence on the probability of detection.

**Figure 5 sensors-25-07298-f005:**
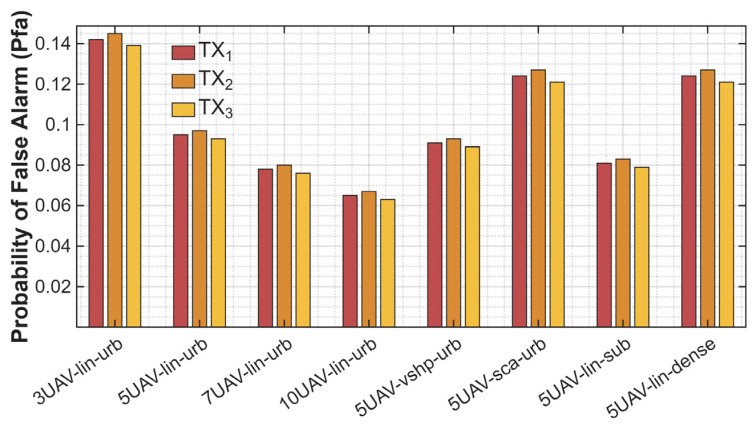
Comparison of scenarios and influence on probability of false alarm.

**Figure 6 sensors-25-07298-f006:**
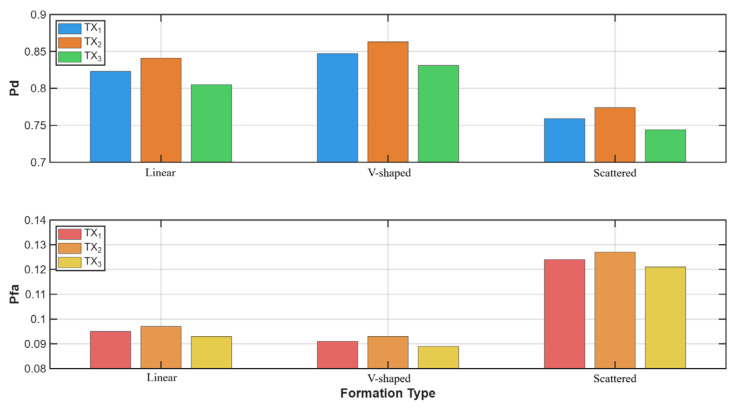
Influence of different formations on probability of detection and false alarms.

**Figure 7 sensors-25-07298-f007:**
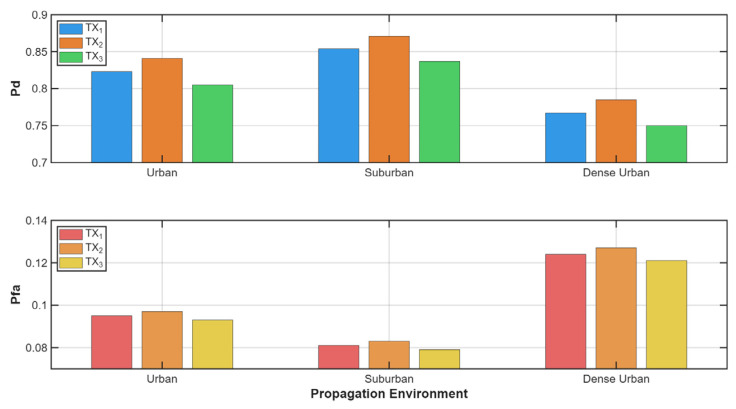
Influence of different environments on probability of detection and false alarms.

**Figure 8 sensors-25-07298-f008:**
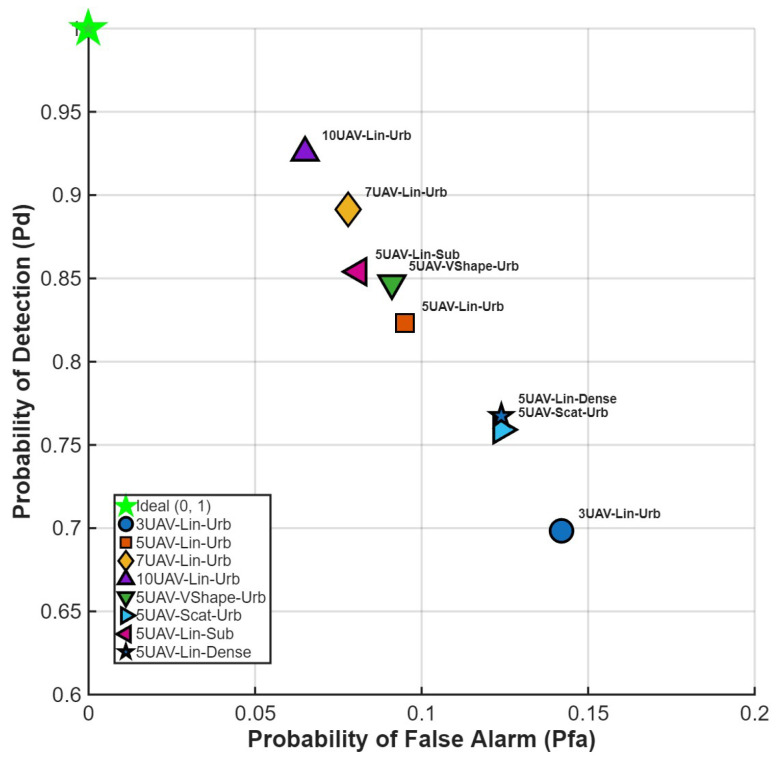
RX operating characteristic for each scenario.

**Figure 9 sensors-25-07298-f009:**
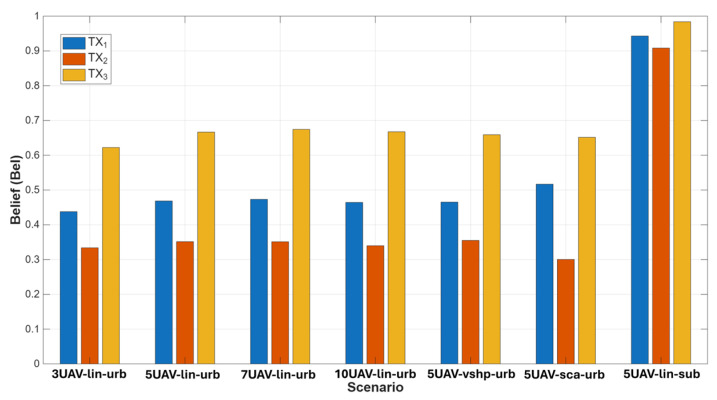
Belief function for different scenarios.

**Figure 10 sensors-25-07298-f010:**
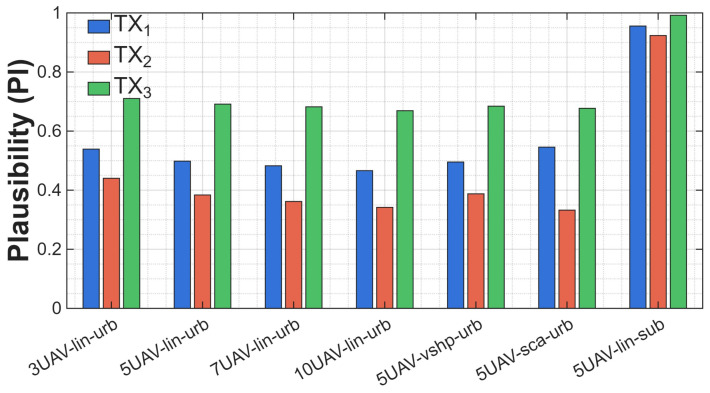
Plausibility function for different scenarios.

**Figure 11 sensors-25-07298-f011:**
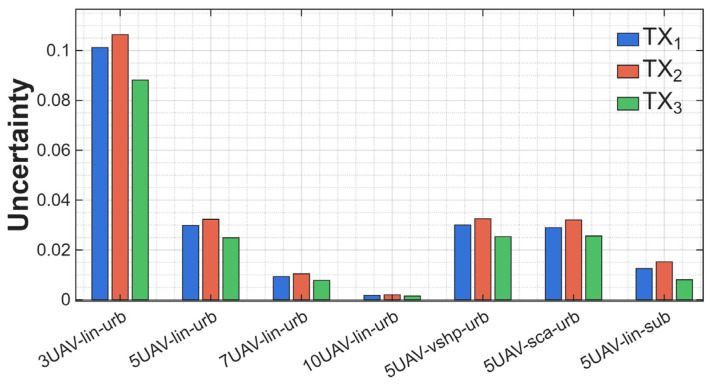
Values of uncertainty of different scenarios.

**Figure 12 sensors-25-07298-f012:**
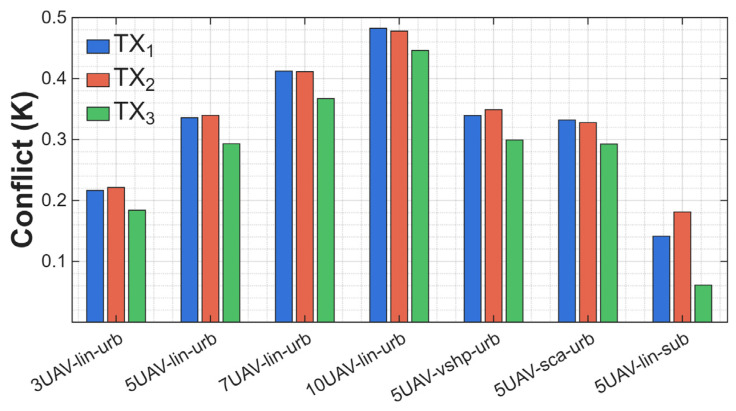
Conflict values of different scenarios.

**Table 1 sensors-25-07298-t001:** Influence of propagation environment for five UAVs in linear formation.

Metrics	Suburban	Urban	Dense
Belief	0.812	0.781	0.724
Plausibility	0.977	0.968	0.955
Uncertainty	0.165	0.187	0.231
Conflict	0.082	0.098	0.145
Pd	0.854	0.823	0.767
Pfa	0.081	0.095	0.124

**Table 2 sensors-25-07298-t002:** Influence of formation with five UAVs and urban environment.

Metrics	Linear	V-Shaped	Scattered
Belief	0.781	0.804	0.723
Plausibility	0.968	0.977	0.961
Uncertainty	0.187	0.173	0.238
Conflict	0.098	0.091	0.142
Pd	0.823	0.847	0.759
Pfa	0.095	0.089	0.132

**Table 3 sensors-25-07298-t003:** Influence of UAV number with linear formation and urban environment.

Number of UAVs	Belief	Uncertainty	Conflict	*P_d_*	*P_fa_*
3	0.642	0.285	0.156	0.698	0.142
5	0.781	0.187	0.098	0.823	0.095
7	0.847	0.124	0.067	0.891	0.078
10	0.892	0.089	0.048	0.925	0.065

## Data Availability

The original contributions presented in this study are included in the article. Further inquiries can be directed to the corresponding author.

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
