# Peer review of "Coordinated Radio Emitter Detection Process Using Group of Unmanned Aerial Vehicles"

_sensors, 2025, doi:10.3390/s25237298_

Round 1

Reviewer 1 Report

Comments and Suggestions for Authors

Major revisions are required.

Coordinated Radio Emitter Detection Process Using Group of Unmanned Aerial Vehicles

The authors tackle the challenges of increasing demand, interference, and the need for reliable adaptive monitoring in the electromagnetic spectrum due to rapid wireless communications expansion by proposing a cooperative monitoring system using groups of UAVs integrated with software-defined radios (SDRs). This approach integrates multiple UAVs for simultaneous energy measurements and data sharing with a fusion center, employs Dempster-Shafer theory (DST) to model uncertainty and combine partial or conflicting sensor data, and utilizes MATLAB-based simulations of swarm formations, mobility, delays, and propagation effects to enhance detection robustness and reduce false alarms. There are a few areas for improvement:

  • In Section 1.3 of the introduction, the contribution point “Data Fusion Approach” mentions that DST provides an “alternative” to traditional hard/soft decision fusion. However, the paper lacks quantitative performance comparisons with these two traditional fusion methods, failing to highlight the superiority of DST.
  • In Section 2.1, when introducing the Energy Detector (ED), it was noted that its computation is straightforward, but its core limitation—a significant performance degradation under low signal-to-noise ratio (SNR) conditions—was not highlighted. This limitation is precisely why multi-UAV coordination is essential. We recommend adding a discussion on the ED's limitations and emphasizing that collaborative sensing and DST fusion are specifically designed to overcome the performance bottlenecks of single-node ED under low SNR and shadowing effects.
  • The hardware combination of USRP B200mini and Raspberry Pi 4 is mentioned, but no explanation is provided for why this combination was chosen. Furthermore, there is no discussion of its processing capabilities, power consumption, or whether it is sufficient to support the 100ms reporting cycle specified in the paper.
  • In Section 2.3, the simulation scenario includes a “moving transmitter” and “varying interference levels,” but the main text fails to describe the specific parameters of the mobile model and interference sources . This results in poor reproducibility of the experiments.
  • In Section 2.4, the identification framework Θ is defined as {H0, H1}, but subsequent BPs are assigned to the subsets {H0}, {H1}, and the entire framework Θ. For a framework with only two mutually exclusive hypotheses, the power set of Θ is {∅, {H0}, {H1}, {H0, H1}}. In the text, m(Θ) denotes “uncertainty,” but it is not explained why {H₀, H₁} can represent uncertainty.
  • In Section 2.4.2, the final decision rule is defined as “Bel(H1) > λ ⇒ Channel is occupied,” but the method for determining the threshold λ remains unspecified. The choice of λ directly impacts Pd and Pfa, making it a core parameter. It is recommended to clarify whether λ is determined by adjusting the ROC curve on the training set or set as a fixed value (e.g., 0.5), and to discuss the rationale for its selection.
  • In Section 3.2, when analyzing the plausibility function (Pl), it was observed that “as the number of sensors increases, the value of the plausibility function decreases,” and this was attributed to “a reduction in the quality assigned to uncertain or alternative hypotheses.” This explanation is unreasonable because Pl(H1) represents the maximum possible quality supporting H1. Typically, as evidence increases, it should become more concentrated on either H1 or H0; Pl(H1) does not necessarily decrease.
  • In Section 3.1.1, the path loss model employs the Okumura-Hata formula, but this model applies to the frequency range of 150 MHz to 1500 MHz and specific antenna heights. The drone height in the paper varies (50–150 meters), while the Okumura-Hata model is sensitive to receiver height. It is not specified whether corrections were made for the drone's airborne receiver.
  • The variables in Equation 1 are not fully explained. Additional clarification of the yk variable could be provided to make the expression of the energy detection method clearer.

10、The conclusion section is relatively comprehensive, but certain parts appear overly general. Specific numerical results could be used to further strengthen the conclusions.

Author Response

Dear Reviewers,

We would like to thank the Reviewers for their insightful comments and suggestions, which significantly contributed to the improvement of our paper. Your constructive feedback enabled us to make valuable revisions that enhanced the clarity and overall quality of our work. We apologize for any lack of clarity or overly brief explanations in some parts of the paper.

In our opinion, thanks to considering the comments and suggestions, the paper is now more readable and substantive. Simultaneously, we hope that the current version of the paper will be accepted, which allow us to publish it in the Sensors.

All changes have been marked with a yellow marker.

Comments 1: [In Section 1.3 of the introduction, the contribution point “Data Fusion Approach” mentions that DST provides an “alternative” to traditional hard/soft decision fusion. However, the paper lacks quantitative performance comparisons with these two traditional fusion methods, failing to highlight the superiority of DST.]

Response 1: [Thank you for this comment. We agree with the Reviewer that performance comparisons were not included in the article. We based our conclusions on previously conducted studies which are now cited.]

Comments 2: [In Section 2.1, when introducing the Energy Detector (ED), it was noted that its computation is straightforward, but its core limitation—a significant performance degradation under low signal-to-noise ratio (SNR) conditions—was not highlighted. This limitation is precisely why multi-UAV coordination is essential. We recommend adding a discussion on the ED's limitations and emphasizing that collaborative sensing and DST fusion are specifically designed to overcome the performance bottlenecks of single-node ED under low SNR and shadowing effects.]

Response 2: [Thank you for this comment. We agree with the Reviewer that limitation of energy detection was not highlighted enough. We have added a deeper analysis of the energy detection limitation]

Comments 3: [The hardware combination of USRP B200mini and Raspberry Pi 4 is mentioned, but no explanation is provided for why this combination was chosen. Furthermore, there is no discussion of its processing capabilities, power consumption, or whether it is sufficient to support the 100ms reporting cycle specified in the paper.]

Response 3: [Thank you for this comment. We agree that the rationale behind selecting the USRP B200mini together with the Raspberry Pi 4 should be explained in more detail. In the revised manuscript, we have expanded the corresponding section to justify this hardware choice.]

Comments 4: [In Section 2.3, the simulation scenario includes a “moving transmitter” and “varying interference levels,” but the main text fails to describe the specific parameters of the mobile model and interference sources . This results in poor reproducibility of the experiments.]

Response 4: [Thank you for this comment. We agree that the specific parameters of the mobile model and interference sources should be explained in more detail. In the revised manuscript, we have expanded the corresponding section to justify this simulation scenario.]

Comments 5: [In Section 2.4, the identification framework Θ is defined as {H0, H1}, but subsequent BPs are assigned to the subsets {H0}, {H1}, and the entire framework Θ. For a framework with only two mutually exclusive hypotheses, the power set of Θ is {∅, {H0}, {H1}, {H0, H1}}. In the text, m(Θ) denotes “uncertainty,” but it is not explained why {H₀, H₁} can represent uncertainty.]

Response 5: [Thank you for this comment. We agree that the interpretation of the framework Θ and its power set requires clarification. In the revised manuscript, we clarify this issue.]

Comments 6: [In Section 2.4.2, the final decision rule is defined as “Bel(H1) > λ ⇒ Channel is occupied,” but the method for determining the threshold λ remains unspecified. The choice of λ directly impacts Pd and Pfa, making it a core parameter. It is recommended to clarify whether λ is determined by adjusting the ROC curve on the training set or set as a fixed value (e.g., 0.5), and to discuss the rationale for its selection.]

Response 6: [Thank you for this comment. We agree that the chosen threshold should be explained in more details. In the revised manuscript, we have expanded the corresponding section to justify this decision and its rationality.]

Comments 7: [In Section 3.2, when analyzing the plausibility function (Pl), it was observed that “as the number of sensors increases, the value of the plausibility function decreases,” and this was attributed to “a reduction in the quality assigned to uncertain or alternative hypotheses.” This explanation is unreasonable because Pl(H1) represents the maximum possible quality supporting H1. Typically, as evidence increases, it should become more concentrated on either H1 or H0; Pl(H1) does not necessarily decrease.]

Response 7: [Thank you for this valuable comment. The reviewer is right that our explanation was imprecise and needed clarification.]

Comments 8: [In Section 3.1.1, the path loss model employs the Okumura-Hata formula, but this model applies to the frequency range of 150 MHz to 1500 MHz and specific antenna heights. The drone height in the paper varies (50–150 meters), while the Okumura-Hata model is sensitive to receiver height. It is not specified whether corrections were made for the drone's airborne receiver.]

Response 8: [Thank you for this comment. We add additional clarification about considering drone’s height.]

Comments 9: [The variables in Equation 1 are not fully explained. Additional clarification of the yk variable could be provided to make the expression of the energy detection method clearer.]

Response 9: [Thank you for this comment. We agree that the definition of the variable yk in Equation 1 requires clarification. In the revised manuscript, we have described it in more detail in the revised version of the manuscript.]

Comments 10: [10、The conclusion section is relatively comprehensive, but certain parts appear overly general. Specific numerical results could be used to further strengthen the conclusions.]

Response 10: [Thank you for your comment. We've made changes to the conclusions section, adding more references to specific results regarding the impact of sensor count and propagation conditions on simulation results.]

Reviewer 2 Report

Comments and Suggestions for Authors

The paper calls: “Coordinated Radio Emitter Detection Process Using Group of Unmanned Aerial Vehicles” and concerned of new approach in monitoring and control aerial wireless communication systems based on set of drones. The advantage of article is unusual approach and long references list (35 items). However reviewer has some questions:

  1. Where real experiment or its just theoretical article (concept)?
  2. Which band do you want use for monitoring (LF, HF or UHF band)? This is for antenna size.
  3. What about downlink channel capacity for transmitting data from sdr? Do you propose use another radio band for downlink transmitting? May be optical fiber links fit for such purpose?
  4. In my opinion such approach can solve problem of radio-interference between different points (receivers) – see fig.2.

Author Response

Dear Reviewers,

We would like to thank the Reviewers for their insightful comments and suggestions, which significantly contributed to the improvement of our paper. Your constructive feedback enabled us to make valuable revisions that enhanced the clarity and overall quality of our work. We apologize for any lack of clarity or overly brief explanations in some parts of the paper.

In our opinion, thanks to considering the comments and suggestions, the paper is now more readable and substantive. Simultaneously, we hope that the current version of the paper will be accepted, which allow us to publish it in the Sensors.

All changes have been marked with a yellow marker.

Comments 1: [Where real experiment or its just theoretical article (concept)?]

Response 1: [Thank you for your comment. For now it’s just theoretical article but in near future we are planning to conduct real experiment with the use of obtained from simulation results.]

Comments 2: [Which band do you want use for monitoring (LF, HF or UHF band)? This is for antenna size.]

Response 2: [Thank you for this comment. We are planning to monitor three types of transmission – radio station (PMR, military communication), base station (LTE, 5G) and UAVs communication.]

Comments 3: [What about downlink channel capacity for transmitting data from sdr? Do you propose use another radio band for downlink transmitting? May be optical fiber links fit for such purpose?]

Response 3: [Thank you for this comment. For real implementation we are planning to use another radio – for example some LoRa module.]

Comments 4: [In my opinion such approach can solve problem of radio-interference between different points (receivers) – see fig.2.]

Response 4: [Thank you for this comment. We agree with the Reviewer that this may be useful in practical implementation.]

Reviewer 3 Report

Comments and Suggestions for Authors

  1. It is suggested that the rationale for selecting Dempster-Shafer Theory (DST) over other fusion methods be more explicitly stated in the introduction to better frame the paper's primary contribution.

  1. To enhance reproducibility, it is recommended that further details regarding the simulation parameters, particularly the specific configuration of the Okumura-Hata channel model, be included.

  1. The clarity of figures with multiple data series (e.g., Figure 4 and Figure 9) could be improved through adjustments to layout, styling, or labeling to make comparisons easier to interpret.

  1. The discussion would be strengthened by a brief mention of the challenges anticipated in a real-world hardware implementation (e.g., synchronization, power constraints), bridging the gap between simulation and practice.

  1. The authors’proposal to integrate machine learning is valuable—we suggest specifying Deep Reinforcement Learning (DRL), which is well-suited for dynamic UAV trajectory and resource optimization. Reference recent relevant studies like "Deep Reinforcement Learning for Energy Efficiency Maximization in RSMA-IRS-Assisted ISAC System" and "Hybrid-RIS-Assisted Cellular ISAC Networks for UAV-Enabled Low-Altitude Economy via Deep Reinforcement Learning with Mixture-of-Experts" to strengthen the outlook.

Author Response

Dear Reviewers,

We would like to thank the Reviewers for their insightful comments and suggestions, which significantly contributed to the improvement of our paper. Your constructive feedback enabled us to make valuable revisions that enhanced the clarity and overall quality of our work. We apologize for any lack of clarity or overly brief explanations in some parts of the paper.

In our opinion, thanks to considering the comments and suggestions, the paper is now more readable and substantive. Simultaneously, we hope that the current version of the paper will be accepted, which allow us to publish it in the Sensors.

All changes have been marked with a yellow marker.

Comments 1: [It is suggested that the rationale for selecting Dempster-Shafer Theory (DST) over other fusion methods be more explicitly stated in the introduction to better frame the paper's primary contribution.]

Response 1: [Thank you for this comment. We agree with the Reviewer that a more thorough justification was lacking. We have expanded the introduction.]

Comments 2: [To enhance reproducibility, it is recommended that further details regarding the simulation parameters, particularly the specific configuration of the Okumura-Hata channel model, be included]

Response 2: [Thank you for this valuable comment. We agree that additional information regarding the simulation parameters is essential for improving reproducibility. In the revised manuscript, we have expanded the methodology section to clearly describe all parameters of the path-loss model used in our simulations]

Comments 3: [The clarity of figures with multiple data series (e.g., Figure 4 and Figure 9) could be improved through adjustments to layout, styling, or labeling to make comparisons easier to interpret]

Response 3: [Thank you for this valuable comment. We have tried our best to improve the clarity of figures and included them in the revised manuscript.]

Comments 4: [The discussion would be strengthened by a brief mention of the challenges anticipated in a real-world hardware implementation (e.g., synchronization, power constraints), bridging the gap between simulation and practice]

Response 4: [Thank you for this valuable suggestion. We agree that addressing the challenges of real-world hardware implementation would strengthen the discussion and better connect the simulation results with practical deployment. In the revised manuscript, we have added a key practical constraints.]

Comments 5: [The authors’proposal to integrate machine learning is valuable—we suggest specifying Deep Reinforcement Learning (DRL), which is well-suited for dynamic UAV trajectory and resource optimization. Reference recent relevant studies like "Deep Reinforcement Learning for Energy Efficiency Maximization in RSMA-IRS-Assisted ISAC System" and "Hybrid-RIS-Assisted Cellular ISAC Networks for UAV-Enabled Low-Altitude Economy via Deep Reinforcement Learning with Mixture-of-Experts" to strengthen the outlook]

Response 5: [Thank you for this valuable comment. We agree that specifying DRL and cite proposed publications may strengthen the outlook.]

Round 2

Reviewer 1 Report

Comments and Suggestions for Authors

The manuscript seems acceptable based on the feedback.